# Sport Participation and Academic Performance in Young Elite Athletes

**DOI:** 10.3390/ijerph192315651

**Published:** 2022-11-25

**Authors:** Tania Pinto-Escalona, Pedro L. Valenzuela, Irene Esteban-Cornejo, Óscar Martínez-de-Quel

**Affiliations:** 1Didactics of Languages, Arts and Physical Education Department, Faculty of Education, Complutense University of Madrid, 28040 Madrid, Spain; 2Physical Activity and Health Research Group (PaHerg), Research Institute of Hospital 12 de Octubre (IMAS12), 28041 Madrid, Spain; 3PROFITH Research Group, Sport and Health University Research Institute (iMUDS), Department of Physical Education and Sports, Faculty of Sport Sciences, University of Granada, 18071 Granada, Spain; 4Centro de Investigación Biomédica en Red Fisiopatología de la Obesidad y Nutrición (CIBERobn), Instituto de Salud Carlos III, 28029 Madrid, Spain; 5Faculty of Sciences for Physical Activity and Sport (INEF), Polytechnic University of Madrid, 28040 Madrid, Spain

**Keywords:** adolescence, youth, sport practice, high-performance athletes, cognitive function, cognitive performance, school marks, school grades, academic achievement

## Abstract

Strong evidence supports physical activity and fitness levels being positively associated with cognitive performance and overall academic performance in youth. This also applies to sports participation. However, whether participation in sports at the elite level is associated with greater academic performance remains unknown. Thus, the present study aimed to compare the academic performance of young elite athletes to that of control students, as well as to analyze whether the type of sport mediates these results. Between 2010 and 2019, all students from the last Baccalaureate course of the Spanish Elite Sport High School—which also includes non-elite athletes and recreational athlete students, who were categorized as controls—participated in this study. Academic performance was assessed through both the grade point average of the two last Baccalaureate courses and through the average grades from the University Entrance Examinations. Athletes were categorized attending to different sport classifications. A total of 1126 adolescents (570 girls, 18.2 ± 0.6 years) participated in the study, of which 483 and 643 were categorized as elite athletes and control students, respectively. Elite athletes attained a lower overall academic performance than controls (*p* < 0.001), which was confirmed for both sexes (*p* < 0.001). These differences were separately confirmed for most academic subjects (*p* < 0.05), as well as when attending to different sport classifications (all *p* > 0.05). Young elite athletes attained a lower academic performance than their non-elite peers, regardless of their type of sport. These findings highlight the importance of programs aimed at facilitating dual careers among young elite athletes.

## 1. Introduction

The benefits of physical activity (PA) and fitness on cognitive performance have been widely recognized [1,2,3,4,5,6]. For instance, Aberg et al. reported a positive association between cardiorespiratory fitness and cognitive performance in adolescents (18 years old) [7]. In the same line, meta-analytical evidence supports a direct association between both PA levels and fitness with academic performance and cognitive performance in children and adolescents [8,9,10], which is in turn related to having a better quality of life and success in the future [11,12]. Indeed, strong evidence supports a beneficial effect of regular PA and exercise on markers of cognitive performance, notably executive functioning, cognitive flexibility, language skills, attention, working memory, or processing speed [1,4,10]. There is also mechanistic evidence supporting a beneficial effect of regular physical exercise—with subsequent increases in cardiorespiratory fitness—on cognition, notably through reductions in anxiety levels, increases in the neuroelectric activity of the cerebral cortex, increases in neurotrophins (e.g., brain-derived neurotrophic factor), and increases in hippocampal blood flow [13,14,15,16,17]. Indeed, higher cardiorespiratory fitness during childhood has been positively associated with the development of distinctive brain regions that are in turn associated with greater academic performance [18].

Preliminary evidence also suggests that sports participation might improve cognitive function and academic performance [19,20,21,22]. For instance, Ishihara et al. recently reported that sports participation was associated with greater academic performance during a two-year follow-up in adolescents aged 12–13 years, which was mediated by the gains in cardiorespiratory fitness [23]. Adolescents participating in team sports presented self-reported higher academic performance than those who did not participate [24]. In the same line, a longitudinal study reported that sports participation predicted better academic performance one year later [25]. Particularly, participating in individual sports or in sports with complex motor skills seemed to be associated with higher school grades [23]. It would therefore be reasonable to hypothesize that young elite athletes, who arguably perform at the greatest PA levels, present the highest cardiorespiratory fitness and achieve the greatest performance level, might attain a greater academic performance than the general population.

Controversy exists on whether sports participation, particularly at the elite level, is actually associated with a greater academic performance [26,27,28]. Elite sports participation has been proposed as a limitation for academic performance [29], and more research is needed to determine whether the positive association between sports participation and academic performance has an upper limit. For example, Becker and colleagues reported a curvilinear association between sport intensity and cognitive function, with sports participation at the highest intensity associated with worse executive functions and academic performance (math scores) [30]. However, a longitudinal study observed that executive functions of high-level soccer players followed the same developmental trajectories as those of the general population despite long-term training [31]. Indeed, Esteban-Cornejo et al. reported that although higher PA levels during early adolescence (11 years) were associated with a greater cognitive performance at 18 years old, higher levels of moderate-to-vigorous PA at 18 years old were associated with an impaired cognitive performance [32]. Importantly, this association may vary depending on the type of sport or sex; however, most of the previous evidence did not differentiate by these variables, and evidence in this regard is still scarce.

The aim of this study was to compare the academic performance of young elite athletes to that of control students, as well as to analyze whether results vary attending to sex and type of sport.

## 2. Materials and Methods

### 2.1. Participants

All students from the last Baccalaureate course (age ~17–18 years) between 2010 and 2019 of the Spanish Elite Sport High School (IES Ortega y Gasset, Madrid, Spain) participated in this study. This is the high school associated to the High-Performance Sport Center in Madrid, where Spanish elite athletes are granted to compete internationally while studying before entering university from the first grade of secondary school to the last Baccalaureate course. Only students from the second course of Baccalaureate participated in this study, since they are the only ones who took the University Entrance Examinations (UEE).

Participants were categorized as either elite athletes (included in the official national list of elite athletes due to their national and international awards) or control students (students from the same high school who may be non-athletes or recreational athletes but who were not included in the official national list of elite athletes). Following the sport classification from Ishihara et al. [23], athletes were categorized into individual and simple sports (swimming, cycling, weightlifting, archery, horse riding, mountaineering and climbing, triathlon, athletics, canoeing, ice skating, golf, orienteering and ski), individual and complex sports (badminton, table tennis, tennis, judo, karate, taekwondo, fencing, wrestling, rhythm gymnastics and artistic gymnastics) and team and complex sports (baseball, basketball, football, volleyball, rowing, rugby, synchronized swimming, hockey, handball and water polo). The study complied with international laws on data protection and with The Code of Ethics of the World Medical Association (Declaration of Helsinki).

### 2.2. Outcomes

Academic performance was assessed through (i) the grade point average (GPA) of the two last Baccalaureate courses (11th and 12th grades) and (ii) the average grades attained of the UEE, both ranging from 0 (lowest score) to 10 (highest score). GPA from the two last Baccalaureate courses was assessed by the same teachers in each academic year, following the evaluation criteria of the Spanish curriculum for each subject.

UEE is a standard evaluation for all students at the regional level, including common exams for Spanish as a native language, English, French or Italian as foreign language, Philosophy or History and a specific subject from their type of Baccalaureate (Technical Drawing, Arts’ History, Biology, Earth and Environmental Sciences, Physics, Mathematics, Chemistry, Business Economics, Geography and Applied Mathematics to Social Science). These exams are developed and evaluated by external teachers; therefore, students are assessed by blinded assessors under unified assessment criteria.

### 2.3. Statistical Analyses

Data are shown as mean ± SD. Normality and homocedasticity were tested using the Kolmogorov–Smirnov and Levene’s test, respectively. Differences between groups were assessed through a one-way analysis of covariance after adjusting for sex, type of Baccalaureate, UEE call and academic year. Sub-analyses were also performed attending to participants’ sex and type of sport. Analyses were performed using SPSS (v25, IBM, Armonk, NY, USA) with α < 0.05.

## 3. Results

A total of 1126 adolescents (570 girls, 18.2 ± 0.6 years) participated in the study, of which 483 were elite athletes (234 girls) and 643 were control students (336 girls). No differences were found between groups for age (*p* = 0.252) or sex (*p* = 0.206).

Elite athletes attained a lower overall academic performance (average of all subjects) than control students when attending to both the Baccalaureate GPA and the UEE examination (*p* < 0.001, Figure 1; Table 1), which was also confirmed in sub-analyses dividing by sex (*p* < 0.001, Table 2). Specifically, elite athletes attained a lower academic performance than control students in all general subjects (i.e., native language, foreign language and history) as well as in most specific subjects (i.e., technical drawing, art’s history, physics, mathematics, chemistry and geography) (Table 1). The results remained essentially the same in sub-analyses attending to different sport classifications (Appendix A).

No significant differences were found in Baccalaureate GPA and the UEE examination (*p* > 0.05) between individual and simple sports, individual and complex sports and team and complex sports (Table 3). No differences (*p* > 0.05) were found either when categorizing athletes attending to other sport classifications. These sport classifications proposed by different authors [33,34,35,36,37,38] were: dynamic or static sport according to its type of intensity (Appendix A); the body muscle groups involved (Appendix A); individual, team or combat sport (Appendix A); acyclic sport, submaximum endurance, upper- and mid-endurance, team sport with high intensity and constant pauses of time, team sport with high intensity and few pauses of time, combat sport or complex sport with multiple test (Appendix A); environment, partner or adversary (Appendix A).

## 4. Discussion

The aim of this study was to compare the academic performance of young elite athletes to that of control students, as well as to analyze whether results vary attending to sex and type of sport. The most important finding was that those individuals who represent the paradigm of the highest level of sports participation (i.e., young elite athletes) attained a worse academic performance than control students regardless of their sport.

Evidence overall suggests that sports participation is associated with greater cognitive performance and academic performance [20,22]. For instance, a longitudinal study showed that adolescents who played sports performed better at English and Mathematics than their classmates who did not participate in any sport [25]. Moreover, in a sample of 6946 adolescents aged 14–17 years, Chen and colleagues found that participating in team sports was associated with a better academic performance, regardless of whether students were involved in one, two, three or more team sports [24]. However, controversy exists as to whether sports participation, particularly at the elite level, is actually associated with a greater academic performance [26,27,29]. 

Indeed, in the present study, we found an inverse association between elite sports participation and academic performance. Barlow and Hickey also found that Division III athletes had lower GPA scores during the competitive season than during the off-season [27]. In this regard, a study conducted among 575 young Spanish elite athletes reported that most of them perceived the combination of their sport with studies as difficult, which was mostly due to time constraints [39]. Similarly, a group of young (under 23 years) elite athletes who competed at the national level reported that they based their subject planning attending to their training schedules, and that it was very difficult for them to attend to their University lessons [40]. Thus, the great amount of time spent doing sport as well as the inflexible academic timetables might be potential reasons underlying the lower academic performance observed in young elite athletes [41]. Efforts are needed to facilitate dual careers among young athletes, which might involve the athletes themselves, educational institutions and sport bodies, the academic staff, the sport staff and the athletes’ families [42]. Moreover, the present findings highlight the need for developing strategies aimed at improving academic performance in young athletes, and although preliminary evidence [43] suggests that psychological skill training (e.g., self-talk, focused attention, goal identification, imagery) could provide beneficial effects, more evidence is needed in this regard.

It could also be hypothesized that the association between elite sports participation and academic performance may rely on the characteristics of each sport. For instance, Ishihara et al. reported that the type of sport (individual vs. team sport, simple vs. complex motor skills) mediates the association between sports participation at the recreational level and academic performance, although these authors found that all types of sports were associated with a greater academic performance [23]. Contrary to these findings, in the present study, we observed that sports participation at the elite level was associated with a worse academic performance regardless of the sports classification used, with no differences between individual and simple sports, individual and complex sports, and team and complex sports. Further research is needed to elucidate how each type of sport affects cognitive performance or academic performance.

Some limitations of the present study should be acknowledged. The cross-sectional design impedes concluding whether sports participation is actually the cause of the lower academic performance found in elite athletes. Moreover, in the present study, we assessed academic performance as an indicator of cognitive function, but no information was available for other important indicators (e.g., executive functions) that are less confounded by variables such as the available time to study, training time, sleeping hours or class attendance, which would have yielded further insights into the actual association between elite sport participation and cognitive performance. Additionally, our results cannot be generalized to other high schools in Spain. In turn, our control group belongs to the same school as the elite athletes’ students, which can be considered a strength, as it would reduce the influence of potential confounding factors. For instance, we have avoided the GPAs attained by the study participants being affected by having different teachers with different methodologies and exams. In turn, the large number of young elite athletes included in the variety of sports and subjects analyzed can be considered a major strength.

## 5. Conclusions

Despite the widely reported association between PA, fitness and sports participation with academic performance [5,15,44], sports participation at the highest competitive level seems to be associated with an impaired academic performance regardless of sex and the type of sport.

## Figures and Tables

**Figure 1 ijerph-19-15651-f001:**
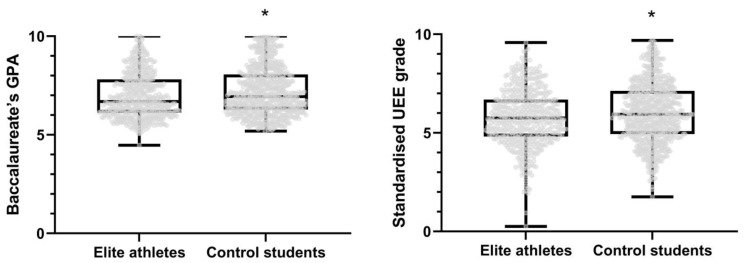
Academic performance (average grade on the University Entrance Examinations [UEE] and the two last Baccalaureate courses) attained by elite athletes and control students. Data represent estimated marginal means after adjusting for sex, type of Baccalaureate UEE call and academic year. * Significant differences between groups (*p* < 0.001).

**Table 1 ijerph-19-15651-t001:** Academic performance of study participants according to sport participation.

	Elite Athletes (*n* = 483)	Control Students (*n* = 643)	*p*-Value
Baccalaureate GPA	6.98 ± 1.1 (*n* = 483)	7.21 ± 1.2 (*n* = 643)	<0.001
Standardized UEE grade	5.76 ± 1.4 (*n* = 483)	6.02 ± 1.5 (*n* = 643)	<0.001
*General UEE subjects*			
Native language (Spanish)	5.89 ± 1.6 (*n* = 483)	6.04 ± 1.7 (*n* = 643)	0.048
Foreign language (English, French or Italian)	6.33 ± 1.9 (*n* = 483)	6.55 ± 2.1 (*n* = 643)	0.006
History/Philosophy	5.45 ± 2.3 (*n* = 483)	5.81 ± 2.3 (*n* = 643)	<0.001
*Specific UEE subjects*			
Technical drawing	5.58 ± 2.1 (*n* = 62)	6.48 ± 2.3 (*n* = 70)	0.045
Arts’ history	3.40 ± 2.4 (*n* = 18)	5.58 ± 2.5 (*n* = 68)	0.003
Biology	5.49 ± 2.6 (*n* = 126)	5.72 ± 2.3 (*n* = 167)	0.545
Earth and Environmental Sciences	5.49 ± 1.9 (*n* = 71)	5.60 ± 1.8 (*n* = 105)	0.345
Physics	4.11 ± 2.3 (*n* = 100)	4.90 ± 2.8 (*n* = 96)	0.024
Mathematics	5.40 ± 2.1 (*n* = 233)	5.78 ± 2.3 (*n* = 219)	0.038
Chemistry	4.10 ± 2.3 (*n* = 119)	4.99 ± 2.6 (*n* = 169)	0.002
Business Economics	5.20 ± 2.1 (*n* = 107)	5.49 ± 2.2 (*n* = 212)	0.197
Geography	4.66 ± 1.8 (*n* = 95)	5.43 ± 1.9 (*n* = 197)	0.002
Applied Mathematics to Social Science	4.43 ± 2.1 (*n* = 150)	4.81 ± 2.7 (*n* = 225)	0.125

Data are shown as mean ± SD. Analyses were adjusted for sex, type of Baccalaureate, UEE call and academic year. Abbreviations: GPA, grade point average; UEE, University Entrance Examinations.

**Table 2 ijerph-19-15651-t002:** Academic performance of study participants according to sport participation by sex.

	Girls (*n* = 570)			Boys (*n* = 556)		
	Control (*n* = 336)	Elite (*n* = 234)	*p*-Value	Control (*n* = 307)	Elite (*n* = 249)	*p*-Value
Baccalaureate GPA	7.43 ± 1.2	7.08 ± 1.2	<0.001	6.97 ± 1.1	6.89 ± 1.1	0.020
Standardized UEE grade	7.52 ± 2.5	6.46 ± 2.1	<0.001	7.14 ± 2.4	6.61 ± 2.3	<0.001

Data are shown as mean ± SD. Analyses were adjusted for type of Baccalaureate, UEE call and academic year. Abbreviations: GPA, grade point average; UEE, University Entrance Examinations.

**Table 3 ijerph-19-15651-t003:** Academic performance of elite athletes based on their type of sport.

	Simple & Individual (*n* = 199)	Complex & Individual (*n* = 133)	Complex & Team (*n* = 151)	*p*-Value
Baccalaureate GPA	7.03 ± 1.1	6.77 ± 1.2	7.08 ± 1.2	0.083
Standardized UEE grade	6.49 ± 2.1	6.40 ± 2.2	6.74 ± 2.2	0.531

Data are shown as mean ± SD. Analyses were adjusted for sex, type of Baccalaureate, UEE call and academic year. Abbreviations: GPA, grade point average; UEE, University Entrance Examinations.

## Data Availability

Data will be made available upon reasonable request to the corresponding author.

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
