# Peer review of "Sport Participation and Academic Performance in Young Elite Athletes"

_ijerph, 2022, doi:10.3390/ijerph192315651_

Round 1

Reviewer 1 Report

Dear all,

The manuscript fits with the aim of the ijerph, and the subject reveals good content for researchers and professionals in the subject of (Effect of Physical Activity, Sedentary, and Fitness on Cognitive Function and Well-Being). However, some points are listed below:

Title

No comments.

Abstract

In Abstract, page1, sentences 25-26: ‘the Spanish Elite Sport High School –which also includes non-athlete students, who were categorized as controls’.

Could you clarify how the Spanish Elite Sport High School has non-athlete students?

Or might you mean (non-elite athletes’ students)?

1. Introduction

It is too short. You can touch a review of similar and related literatures, grade point average (GPA), University Entrance Examinations (UEE), important attendance, and behaviours.

2. Materials and Methods

Page2, sentences 72: ‘All students from the last Baccalaureate’s course (age ~17-18 years) between 2010 and 2019 of the Spanish Elite Sport High School’

Could you clarify why you chose only the Baccalaureate’s level?

Did the other stages have not any elite players?

3. Results

Detecting academic performance for elite athletes without considering the attendance rate (elite athletes attendance rate compared with non-elite athletes) will hide many results that are not in favour of the elite athletes compared to controls who attended more classes.

There are other experimental factors that should have been studied and controlled (e.g. training hours - sleeping hours - study hours per day - compensation for missed lectures) and in order to adjust the results.

4. Discussion

Please start the Discussion with a short sentence explains the study aim, the most important findings of the study like “The most important finding of the present study was…”. 

References

References are not adequate to IJERPH style. For example:

Journal Articles:

1. Author 1, A.B.; Author 2, C.D. Title of the article. Abbreviated Journal Name (italic) Year (bold)Volume, page range.

Please check all references be compatible with IJERPH style (ACS style).

Author Response

[Response] We sincerely appreciate the Reviewers’ comments. We have revised our manuscript in depth following each of their recommendations. Changes are tracked (red font) in the revised version. Please find below our point-by-point responses.

Reviewer 1

Dear all,

The manuscript fits with the aim of the IJERPH, and the subject reveals good content for researchers and professionals in the subject of (Effect of Physical Activity, Sedentary, and Fitness on Cognitive Function and Well-Being). However, some points are listed below:

Comments much appreciated. Regarding the limitations, please find below our point-by-point responses and the changes made on the manuscript. We believe we have amended or explained all limitations in the revised version.

Title:

No comments.

Abstract:

In Abstract, page1, sentences 25-26: ‘the Spanish Elite Sport High School –which also includes non-athlete students, who were categorized as controls’.

Could you clarify how the Spanish Elite Sport High School has non-athlete students?

Or might you mean (non-elite athletes’ students)?

We agree with the reviewer comment. The Spanish Elite Sport High School includes elite athletes and students who are not included in the official national list of elite athletes. Thus, controls can be either non-athletes, or participate in sports but at the recreational level. For the sake of simplicity and given the word constraints imposed for the abstract, we have just specified “non-elite athletes and recreational athletes´ students” (see page 1, line 8). However, in the Methods section of the manuscript we have explained the control group in detail. “Participants were categorized as either elite athletes (included in the official national list of elite athletes due to their national and international awards) or control students (students from the same high school who may be non-athletes or recreational athletes but who were not included in the official national list of elite athletes)” (see page 3, lines 90-93).

  1. Introduction

It is too short. You can touch a review of similar and related literatures, grade point average (GPA), University Entrance Examinations (UEE), important attendance, and behaviours.

We have expanded the introduction section including more literature regarding the benefits of physical activity, exercise and sport on cognitive functioning and academic performance (see page 2).

  1. Materials and Methods

Page2, sentences 72: ‘All students from the last Baccalaureate’s course (age ~17-18 years) between 2010 and 2019 of the Spanish Elite Sport High School’

Could you clarify why you chose only the Baccalaureate’s level?

Did the other stages have not any elite players?

We appreciate the Reviewer’s suggestion.  We have provided further information about the participants in the revised manuscript: “from the first grade of secondary school to the last Baccalaureate’s course. Only students from the second course of Baccalaureate participated in this study, since they are the only ones who took the University Entrance Examinations (UEE)” (see page 3, lines 86-89).

The Spanish Elite Sport High School has elite students in all the Secondary and Baccalaureates’ courses. However, we decided to include those who belonged to the last Baccalaureate course since they are the only ones who attended to the University Entrance Examinations (UEE). We believe that UEE is the most objective examination since an external and anonymous examination is performed, so we think it could be seen as a “gold standard” for academic performance.

  1. Results

Detecting academic performance for elite athletes without considering the attendance rate (elite athletes attendance rate compared with non-elite athletes) will hide many results that are not in favour of the elite athletes compared to controls who attended more classes.

There are other experimental factors that should have been studied and controlled (e.g. training hours - sleeping hours - study hours per day - compensation for missed lectures) and in order to adjust the results.

We agree with the Reviewer that both the attendance variable and the other variables proposed (e.g. training hours - sleeping hours - study hours per day - compensation for missed lectures) should have been taken into account in a controlled way in order to adjust the results, however we do not have this information in our study. Thus, it has been now specified in the limitations section: “no information was available for other important indicators (e.g., executive functions) that are less confounded by variables such as the available time to study, training time, sleeping hours or classes’ attendance, which would have yielded further insights into the actual association between elite sport participation and cognitive performance” (see page 7, lines 195-199)

  1. Discussion

Please start the Discussion with a short sentence explains the study aim, the most important findings of the study like “The most important finding of the present study was…”. 

Done. “The aim of this study was to compare the academic performance of young elite athletes to that of control students, as well as to analyze whether results vary attending to sex and type of sport. Therefore, the most important finding was that those individuals who represent the paradigm of the highest level of sport participation (i.e., young elite athletes) attained a worse academic performance than control students regardless of their sport” (see page 6, lines 148-153).

References

References are not adequate to IJERPH style. For example:

Journal Articles:

  1. Author 1, A.B.; Author 2, C.D. Title of the article. Abbreviated Journal Name (italic)Year (bold)Volume, page range.

Please check all references be compatible with IJERPH style (ACS style).

Thanks, now all the references are in the ACS style.

Reviewer 2 Report

Thank you for the opportunity to review this manuscript. In my opinion, this is a quality study that provides original findings and has a potential to make an important contribution to clarifying the relationship between academic achievement and participation in sports. I also appreciate the detailed analysis of the academic achievement of athletes participating in different sports sorted according to different classifications.

Before I can recommend the article for publication, however, a few points need to be clarified.

(i) The definition of the control group is very confusing. The abstract and the introduction (line 64) imply that these are "non-athlete" students. However, in the materials and methods section (line 79) it states that these are "non-elite athletes". The definition of the control group needs to be clarified. I believe it is crucial to define whether these are students who are not involved in sports or athletes who are not elite. This can quite fundamentally affect the interpretation of your results. A conclusion in the abstract referring to the general population could then be misleading.

(ii) The results of the study lack a broader context. We learn that academic performance may be lower in elite athletes than in the control group. But how do the two groups studied perform in the context of other Spanish schools? At the very least, a comparison of this school's results with standardized UEE results at other schools in Spain would be in order. I would highly recommend providing this context in the discussion of the article. In other words: Is the inferior academic performance of elite athletes really inferior in the context of the Spanish education system, or can it only be considered inferior in the context of this one school?

(iii) I think it would be useful to expand the Introduction section. I would recommend moving some passages from the Discussion section here (see the comment to the discussion). I would recommend that the authors conduct a more thorough literature search. They state that to the best of their knowledge, no previous study has compared the academic performance of young elite athletes to that of non-athlete students. However, a quick search reveals that at least one such study exists: https://doi.org/10.1080/13573322.2021.1919070. Authors may also consider other studies: https://doi.org/10.5993/AJHB.41.2.9; https://doi.org/10.1371/journal.pone.0253142; https://doi.org/10.3389/fpubh.2021.730497

(iv) There is a discrepancy between the number of girls listed in the text of the results: 570 (line 110) and Table 2: 571 (page 5, lines 162-163). Please check and correct the numbers of participants.

(v) I think the Discussion section needs to be reworked. I have a few comments on it: (a) Please delete the first paragraph (lines 172-175). It looks more like an instruction for writing this chapter than the actual text of the discussion. (b) The second paragraph is too general and misleading. In the context of this study, it is necessary to distinguish between "individuals participating in sports" and "elite athletes". Similarly, it is necessary to distinguish between “general population” and “non-athletes/non-elite athletes”. Please clearly define the control group for the entire manuscript and then refine the text of the discussion. (c) A significant part of the text of the discussion should be moved to the introduction section (for example, paragraphs 3 and 4, lines 179-209). This is because this text presents findings from other studies (which should have been presented in the introduction) rather than attempting to interpret the results of this study and place them in the context of current knowledge. (d) If you wish to include the studies I mentioned in (iii), or any others you may find, in the introduction section, I think it would be desirable to comment on your results in relation to these studies in the discussion section as well.

(vi) I believe the references in the conclusions section are redundant (line 252). Please consider removing them. Although I agree with your sentence "Given the importance of promoting athletes' development beyond sports, efforts are needed to facilitate dual careers among elite athletes" (lines 254-255), I believe it should not be included here. The results of this study do not directly support this claim (the study did not verify that dual careers lead to better academic performance). It would probably be better to just mention this in the discussion.

Minor comments:

- In the Institutional Review Board Statement (page 11, line 319), there is redundant repetition of “by the”.

- Please check if the Informed Consent Statement (page 11, lines 321-322) is applicable for your study. Was it necessary to request informed consent from all 1126 participants? Or was the consent provided only by the school administration.

- References should have the year of publication in bold, not the volume of the journal.

Author Response

[Response] We sincerely appreciate the Reviewers’ comments. We have revised our manuscript in depth following each of their recommendations. Changes are tracked (red font) in the revised version. Please find below our point-by-point responses.

Reviewer 2

Thank you for the opportunity to review this manuscript. In my opinion, this is a quality study that provides original findings and has a potential to make an important contribution to clarifying the relationship between academic achievement and participation in sports. I also appreciate the detailed analysis of the academic achievement of athletes participating in different sports sorted according to different classifications. Before I can recommend the article for publication, however, a few points need to be clarified.

Thanks for your comments. Please find below our point-by-point responses and the changes made on the manuscript. We believe we have amended or explained all limitations in the revised version.

(i) The definition of the control group is very confusing. The abstract and the introduction (line 64) imply that these are "non-athlete" students. However, in the materials and methods section (line 79) it states that these are "non-elite athletes". The definition of the control group needs to be clarified. I believe it is crucial to define whether these are students who are not involved in sports or athletes who are not elite. This can quite fundamentally affect the interpretation of your results. A conclusion in the abstract referring to the general population could then be misleading.

We agree with the reviewer’s comment and we have now specified in the methods section that the control group includes students from the same school that could be recreational athletes or non-athletes: “control students (students from the same high school who may be non-athletes or recreational athletes but who were not included in the official national list of elite athletes)” (see page 3, lines 91-93).

(ii) The results of the study lack a broader context. We learn that academic performance may be lower in elite athletes than in the control group. But how do the two groups studied perform in the context of other Spanish schools? At the very least, a comparison of this school's results with standardized UEE results at other schools in Spain would be in order. I would highly recommend providing this context in the discussion of the article. In other words: Is the inferior academic performance of elite athletes really inferior in the context of the Spanish education system, or can it only be considered inferior in the context of this one school?

 Thanks for the comment. We decided to include the control group from the same school in order to avoid that the academic results were influenced by having different teachers with different methodologies and exams. However, we have included as a limitation in the discussion section that the results obtained are not generalizable to other schools in Spain: “Moreover, our results cannot be generalized to other High Schools in Spain” (see page 7, lines 199-200). Unfortunately, we have no access to UEE results of other schools in Spain.

(iii) I think it would be useful to expand the Introduction section. I would recommend moving some passages from the Discussion section here (see the comment to the discussion). I would recommend that the authors conduct a more thorough literature search. They state that to the best of their knowledge, no previous study has compared the academic performance of young elite athletes to that of non-athlete students. However, a quick search reveals that at least one such study exists: https://doi.org/10.1080/13573322.2021.1919070. Authors may also consider other studies: https://doi.org/10.5993/AJHB.41.2.9; https://doi.org/10.1371/journal.pone.0253142; https://doi.org/10.3389/fpubh.2021.730497

Thank you for your comment and for the references, which we have read and included in the manuscript (see references 24, 25, 28 and 29). We have also expanded the introduction section, as suggested by the reviewer (see page 1). Moreover, we have deleted that to the best of our knowledge, no previous study has compared the academic performance between elite athletes and non-athletes students.  

(iv) There is a discrepancy between the number of girls listed in the text of the results: 570 (line 110) and Table 2: 571 (page 5, lines 165-166). Please check and correct the numbers of participants.

Thank you for detecting this mistake. The correct number of girls is 570. It is now corrected in Table 2.

(v) I think the Discussion section needs to be reworked. I have a few comments on it: (a) Please delete the first paragraph (lines 172-175). It looks more like an instruction for writing this chapter than the actual text of the discussion. (b) The second paragraph is too general and misleading. In the context of this study, it is necessary to distinguish between "individuals participating in sports" and "elite athletes". Similarly, it is necessary to distinguish between “general population” and “non-athletes/non-elite athletes”. Please clearly define the control group for the entire manuscript and then refine the text of the discussion. (c) A significant part of the text of the discussion should be moved to the introduction section (for example, paragraphs 3 and 4, lines 179-209). This is because this text presents findings from other studies (which should have been presented in the introduction) rather than attempting to interpret the results of this study and place them in the context of current knowledge. (d) If you wish to include the studies I mentioned in (iii), or any others you may find, in the introduction section, I think it would be desirable to comment on your results in relation to these studies in the discussion section as well.

Thanks for your comments. (a) The first paragraph belonged to the journal's instructions for this section and by mistake when we adapted the article to the template we did not delete it but now it is done (see page 6, line 148). (b) We have rewritten the first paragraph by specifying that elite athletes had worse academic performance than control students: “Therefore, the most important finding was that those individuals who represent the paradigm of the highest level of sport participation (i.e., young elite athletes) attained a worse academic performance than control students regardless of their sport” (see page 6, lines 150-153). Moreover, we have defined the control group in the methods section, as mentioned above. (c and d) Paragraphs 3 and 4 have been moved to the introduction section and some studies mentioned in the introduction section were also included in the discussion.  

(vi) I believe the references in the conclusions section are redundant (line 252). Please consider removing them. Although I agree with your sentence "Given the importance of promoting athletes' development beyond sports, efforts are needed to facilitate dual careers among elite athletes" (lines 254-255), I believe it should not be included here. The results of this study do not directly support this claim (the study did not verify that dual careers lead to better academic performance). It would probably be better to just mention this in the discussion.

Following the Reviewer’s suggestion, we have removed the references. Moreover, now the sentence of dual careers is only in the discussion section.

Minor comments:

- In the Institutional Review Board Statement (page 11, line 319), there is redundant repetition of “by the”.

Thanks, we have deleted one “by the”.

- Please check if the Informed Consent Statement (page 11, lines 321-322) is applicable for your study. Was it necessary to request informed consent from all 1126 participants? Or was the consent provided only by the school administration.

We agree with the reviewer and we have changed this section.

- References should have the year of publication in bold, not the volume of the journal.

Thanks, now the references are in the ACS style

Round 2

Reviewer 2 Report

I would like to thank the authors for considering all my recommendations. I am pleased that all the shortcomings I pointed out in the previous round of peer review have been addressed. I am now happy to recommend the manuscript for publication.

While I understand that the authors do not have access to the standardized UEE results of other schools in Spain, as a reader I would still be very interested to know how the two groups of students at the chosen school compare (in their academic performance) to those of other educational institutions or the national average. Perhaps this could be a topic for further research studies.

Author Response

Thank you very much for your comments. As you suggest, we will consider the national average results  in future research studies.